# Did Emotional Intelligence Traits Mitigate COVID-19 Uncertainty Effects on Financial Institutions' Board Decision-Making Process?

Jessica Hall [1], Gregory Jones [2], Claire Beattie [2,3] and John Sands [2,*]

[1] AustralianSuper, GPO Box 1901, Melbourne 3001, Australia; jessicahall@australiansuper.com
[2] School of Business, University of Southern Queensland, Toowoomba 4352, Australia; gregory.jones@unisq.edu.au (G.J.); claire.beattie@lincoln.ac.nz (C.B.)
[3] Department of Financial and Business Systems, Lincoln University, Selwyn, Canterbury 7647, New Zealand
[*] Correspondence: john.sands@unisq.edu.au; Tel.: +61-7-46315385

**Abstract:** This study uses a qualitative research mixed methods design to explore the Coronavirus pandemic's uncertainty effect on mature board governance practices and a board decision processes framework within 16 large Australian financial services entities. Findings provide support for two effects. Firstly, the Coronavirus pandemic had led to a hesitation effect on the board members on-going journey of developing a conscious sense of 'self' and awareness. Secondly, the skills and diversity of personalities of directors comprising the board has a positive impact on the effectiveness and success of strategic decisions. The ongoing ambiguity impact of the Coronavirus pandemic on effective board decision-making processes was investigated. The board members expressed confidence in the Australian financial services sector's ability to overcome the global Coronavirus pandemic's temporary uncertainty impact on board decision processes frameworks. Future research may extend the focus to senior executives' or owners' EI personality traits to investigate the relationship between such individual's or teams' traits and ongoing effective board decision-making processes during uncertainty in either developing or developed countries or a cross-cultural study.

**Keywords:** Coronavirus pandemic; COVID-19; bank director; bank board; decision processes; effectiveness

## 1. Introduction

Since March 2020, Australia and the world have faced the Coronavirus global health pandemic which challenged all global economies and financial institutions in an unprecedented manner during this period. These challenges for financial institutions follow on from regulatory pressures related to the use of their equity reserves as buffers to "absorb losses and sustain the flow of credit to the broader economy" (Frost and Eyers 2020). The literature has identified that emotional intelligence (EI) plays an important role in an organization's effectiveness (Srivastava 2013). A South African-based study found that EI traits provide a significant enhancement to the positive coping behaviour capabilities of individuals (du Plessis 2023). From a study in China, Prentice et al. (2020) found that EI plays an important role in developing positive coping strategies. Furthermore, a study of Lebanese participants demonstrated that EI showed a significant positive effect on the intuitive decision-making style (El Othman et al. 2020). Findings from a study in India are consistent with the concept that EI prepared individuals to "better to deal with acute stressful situations like COVID-10" (Bhatt and Sharma 2020, p. 36). Bhatt and Sharma (2020, p. 33) concluded that "...there is clearly need for more focussed studies looking at correlation of EI with this specific stress situation...". The Lebanese-based study's findings of Sanchez-Ruiz et al. (2021) provide an increasing body of evidence about the benefits of the EI trait and its implications, especially trainings designed to target emotional competencies

as a meaning-centred coping mechanism for individuals in stressful situations during the pandemic.

Analysts have been making inquiries to bank executives and boards about their decision-making relating to how low equity levels will be allowed to drop and what impact this will have on dividend decisions. This optimistic approach was reported to be an appropriate prudent decision-making strategies approach by the board of financial institutions (Frost and Eyers 2020). These authors reported that the appropriate decision-making strategies by the board include an expected (1) decrease in dividend from the current COVID-19 crisis, which is hoped to be temporary and (2) maintenance of the boards' governance responsibilities.

There is some global evidence about the benefits of EI traits, including during the presence of COVID-19. However, no study for the Australian environment was identified in the literature. The Australian Stock Exchange (ASX) reported that the financial sector represented 33.5% of its top 100 companies in 2019 (ASX 2023). Therefore, an investigative study is warranted into these issues for the Australian financial institutions. Therefore, the following research question was developed for this study: Is the possession of EI traits by board members associated with an appropriate prudent decision-making strategies approach taken by the board to maintain the boards' governance responsibilities during these uncertain times? This paper has been developed using the following structure. The next section provides a background on the governance practices and decision-making processes. A discussion in the Section 3 relates to the research method and qualitative data analysis whereby Nvivo version 12 software was used to produce the findings of the study. The final section outlines the conclusions, limitations, and implications for further research.

## 2. Background and Literature Review of Board Decision-Making Processes

A systematic literature review methodology employing six steps was used to review and critically evaluated the relevant theory relating to the overarching research question. After selecting the topic for this study, the literature search was the second step, which focussed on emotional intelligence (EI) personality trait theory (Petrides and Furnham 2001), implicit theories of intelligence (Yeager and Dweck 2012) within the five-factor model of personality traits developed by (Cattell 1943), and associated studies. The first of the next four steps was used to critique the literature related to EI traits, the ASX corporate governance practices related to the Australian financial institutions, and the effect of EI traits coping mechanisms on the decision-making process in times of uncertainty surrounding the COVID-19 pandemic. Step 3 aided the remaining three steps toward developing an informed understanding of researchers' views, the identification of gaps in the existing body of academic knowledge and helped form an argument that is the basis for the research question to serve as the means for a contribution of new knowledge from this research.

In this section, the background discussions about EI's traits and characteristics related to corporate governance and decision-making processes are identified under separate sections and sub-sections.

### 2.1. EI Traits and Effective Board Governance

The five-factor model of personality traits developed by (Cattell 1943) has been refined several times down to a two-factor model referred to as the hierarchy of personality trait theories, which refer to the General Factor of Personality (the g-factor within the GFP model) (Musek 2007). The 'g factor' measures personality factors of emotionality, self-esteem, and personal well-being. These EI personality traits are considered to be behavioural personal traits that can be learned and developed over time rather than static (Goleman 1995). Petrides (2009) developed a measurement of EI Personality Traits through self-reporting. Intelligence is often referred to as an ability or trait of personality "that moves you in a practical and effective way in the world" (Peterson 2017, p. 1). A study by van der Linden et al. (2017) found a sufficiently close correlation between trait EI and the g-factor which is an interchangeable measure of the self-reported emotional intelligence traits of an

individual. The EI traits research literature review supports the proposition that the higher an individual's progress in an organisation, the greater the associated value of behaviours that utilise personality traits to manage emotions effectively and interact socially with others (managing people) increases. Therefore, for board directors, emotional intelligence is critical at the board level (Dulewicz and Higgs 2003).

The complex and highly regulated financial services industry in Australia continually presents challenges and risks for boards. Boards are expected to govern effectively and responsibly, by acting in the best interests of the company, or in the best interests of members of trustee boards of the Responsible Service Entity ('RSE') licensees, and "a high performing, effective board is essential for the proper governance of an entity" (ASX Corporate Governance Council 2014). In 2019, both APRA and ASIC, two of Australia's independent external regulators responsible for monitoring Australia's largest financial banks and superannuation funds, introduced new initiatives into these entities' boardrooms in response to the Royal Commission's (Hayne 2019) investigations into misconduct in the banking, superannuation, and financial services industries. This report showed a severe level of misuse of the trust and confidence placed by everyday Australians in Australia's largest financial institutions (Hayne 2019). Scrutiny from the Royal Commission highlighted that a board's culture and responsibilities encompass a high level of morality, ethics, social responsibility and expectations of community standards, and the need to be fully informed to make independent, unbiased, and cohesive decisions. It was also found that there was a pervasive and systematic failure of boards and executive management to lead with a culture that focussed on the best interests of their customers, members, or shareholders. The consequences of this report had the potential to severely undermine the Australian community's expectations and trust in all of the well-known financial institutions and reputations of these organisations, including the boards, to do the right thing, obey the law, and act ethically.

These regulators considered that whilst there was no exact model in terms of the decision-making process and governing as a functioning board:

> *when you think about . . . what sort of questions that boards are asking, how they're performing their oversight functions . . . there are some . . . good practices that we might identify or . . .where people can improve.* (Durkin 2019, p. 1)

Therefore, boards need to be able to adapt to changing environments, look for improvements, and modify their actions as required. This requires boards to possess or acquire the skills and knowledge, and particularly the mindset, to be able to modify their actions as needed.

The expectations on boards consist of more than just the requirements stipulated in the Australian legislative framework of director's duties in the *Corporations Act 2001* (Cth), the *Superannuation Industry Supervisory Act 1993* (Cth) for trustee directors, and the prudential regulations and guidance issued by the external regulators. Australian boards must also consider the Australian Stock Exchange (ASX) Corporate Governance Council's Corporate Governance Principles and Recommendations when performing their duties. For listed ASX companies, which include Australia's largest banks, insurers, and fund managers, this disclosure is on an "if not, why not" basis. If the companies are not complying with the principles, then they are required to provide an explanation of "why not".

### 2.2. Board EI Trait Skillsets on Effective Group Decision-Making

The relationships between higher-order EI personality factors are similar between men and women (Siegling et al. 2015). Individuals can develop a sophisticated level of knowledge over time to utilise skills and personality behaviours (EI). This can help serve as a framework for approaching one's behaviour of oneself and others' behaviour (Mayer et al. 2002, 2008).

A greater awareness of one's trait emotional intelligence helps an individual identify where they tend to function on personality spectrums in terms of the lower order emotion related personality traits. Where individuals seek to understand and become more aware

of this, it is likely to enable individuals to be more informed to develop changes in their behaviour that increases their tendency to use emotional intelligence personality traits in social interactions in effective ways. This in turn could also assist individuals in developing and extending their personality trait 'toolkit' of emotion related facets and employ more variety in their behaviours. This is understood to lead to a greater movement across the higher order personality traits, and therefore help improve how individuals behave and interact in socially effective ways.

The relevance of emotions in decision-making processes and the relationship between EI traits and decision making has been researched and well recognised (Sevdalis et al. 2007; Fabio and Palazzeschi 2009). One of the beneficial outcomes identified was having a more cohesive and effective way of leading together towards a common purpose (Parrish 2015).

The Australian financial services sector operates in a knowledge-based economy and Directors of Australia's largest financial institutions must navigate through significant uncertainty and risk. Consequently, the primary role of the directors is to make good quality decisions in these conditions (Milkman et al. 2009). A board is unable to manage all extrinsic variables whilst navigating through significant economic risks. In order to maintain the integrity and stability of Australia's financial services system, the "identification and management of human variables [on the board] such as emotion and logic are pivotal in the effort to increase the quality of decisions and decision-making processes" (Hess and Bacigalupo 2013). As part of a board chair's toolkit, an aggregated board director skill set matrix (Adams et al. 2018) supports the chairperson's function of selecting a capable board to lead the strategy of the organisation over the long term. The board skill set matrix published by Australia's largest top 200 ASX listed entities covers four key areas: (1) industry knowledge; (2) technical skills and experience; (3) governance competencies; and (4) behavioural competencies (Kiel et al. 2012). Within the behavioural competencies, the skills that can be usually cited across the varying board skills matrix include collaboration, listening skills, verbal communication, understanding of effective decision-making processes, common sense, ability and willingness to challenge, interpersonal relations, integrity, and mentoring capabilities (Kiel et al. 2012).

Group decision-making can either be effective or ineffective depending on how freely members of the group are able to express their opinions and the robustness of discussions. Hirokawa and Rost (1992) established findings based on a critical examination of vigilant interaction theory which posited that the quality of the group as a decision-making team is dependent upon the group's attentiveness during interaction. Rigorous discussion and critical thinking at all stages of decision making are important to group members in order to produce an optimal decision (Papa et al. 2008). This is carried out by directors with a view to improving the quality and vigilance of board discussion and decision-making processes, thus enabling greater board governance effectiveness in decision-making (Petri and Soublin 2010).

Whether a board chair of an Australian financial services entity, such as a large superannuation fund or bank, has facilitated effective board decision-making between the directors which have formulated a team design and organisational design fit for purpose in achieving the organisation's strategy can be measured in two ways. Internally, through board performance assessments and externally, based on the output of decisions (Leblanc 2016). A key determinant of whether board decision-making is effective can be assessed using the responses of external regulators and its shareholders, particularly the largest institutional shareholders, namely the APRA regulated superannuation funds, insurance companies and fund managers (Institutional Analysis 2010), and proxy voters (Ramsay et al. 2010). Whether or not a chair of a trustee board of a large Australian superannuation fund has facilitated effective board governance can be determined by the responses from its members, rating agencies, and external regulators.

The intended purpose of this literature review is to provide value to boardroom decisions with a view that this will make a meaningful and positive impact on the change required in the culture of some large Australian financial institutional entities. However, it

is more important to add a level of additional resilience and empathy in board decision-making processes to uphold the confidence and trust in Australia's boards of institutions operating in its financial system into the future. Sheedy and Griffin (2017) noted that much of the research conducted on risk governance and board behaviours focussed exclusively on external measures rather than on examining the internal risk governance practices of the board. This research is relevant to the Australian financial service industry's boards and directors facing these challenges. The period of this study presents a time when the boards and management of organisations in Australia's financial system have the opportunity to look inward at the mindset of directors, the culture of the board, and the effectiveness of decision-making processes. The outcome of this research also aims to have practical value in terms of presenting an additional lens on the proper governance practices of Australia's financial services institutions by furthering the field of board decision-making processes. That is, through exploring whether directors are challenging each other and management with vigilant robust discussion during board decision-making processes has enabled an original contribution of knowledge into this academic area of research. Directors' decision-making processes should be viewed as a valued asset to organisations that contribute to good governance practices.

The timeliness of this study should have allowed board members 14 months to familiarise themselves with the requirements of the Royal Commission. However, Australia and the world faced the Coronavirus global health pandemic which challenged all global economies and financial institutions in an unprecedented manner. This context, in itself, accentuated the circumstances of the research.

### 3. Research Method and Data Analysis

This research focuses on the effect of EI traits on the board members' decision-making about governance practices during the COVID-19 years for Australia's large financial institutions. In Australia, there are 17 banks and 445 organisations offering financial services but not all these organisations are large financial institutions. The Australian financial services sector, and the characteristics of their relevance to the research question, was based on providing a sufficiently diverse set of views in the case study. The rationale for selecting a sample size of 18 was a result of the challenge of accessing the elite group of directors on the boards of Australia's financial service entities. A network selection approach was used for seven directors while the remaining 11 directors volunteered using a referral snowballing approach. The data collected were reliable, valid, and based on the relevant units of analysis (Yin 1994).

The case study was conducted over the six-month period, with each of the 18 participating directors who volunteered to participant and took place between January and June 2020. Of the 18 directors, the profile comprised 78% females, 22% males, 50% of which were chairs of their board or committee, and the remaining 50% were only board members (measured in December 2019 before the case study commenced). The organisational mix of the 16 Australian financial services entities was 11% insurance, 33% superannuation (both industry and retail superannuation funds), 28% banking, and 28% wealth management. Of the 16 Australian financial services entities, 7 were listed on the ASX. Collectively, the 16 entities had control and oversight of more than $620 billion in funds under management and market capital (based on 30 June 2019 financial reports).

The Coronavirus global health pandemic challenged all financial institutions in an unprecedented manner. This context, in itself, accentuated the circumstances of the research but it was considered appropriate to provide richer insights from the directors. This was considered to be the case by the directors as vigilant discussion and constructive debate enabled differing views to be shared by the directors and enabled directors to be persuaded otherwise by the supporting views and alternative options presented.

The data collection methods described were congruent with addressing the research question and applying a sequential mixed method methodology in the quantitative–qualitative research design approach. There was a 100% successful participation rate

achieved for the surveys (quantitative design) conducted before and after the interviews (qualitative design). A period of 6 months intervened between the first survey and the second survey. This period permitted directors to be interviewed and provided them with an opportunity to use the information in the surveys in their decision-making roles at meetings. A 100% successful participation rate was achieved for the interviews. An interview generally spanned between 30 minutes to an hour, depending on how much discussion was generated to cover the 20 interview questions. The interview participants included independent directors, board chairs, and executive managing directors of Australian financial services institutions. The participants voluntarily participated in the case study, were unrelated, and did not necessarily know each other. These methods have allowed the research data and information to be collected in a manner that is measurable and meaningful.

Vigilant interactive theory was used in the context of effective board governance practices. This served as the basis for developing an appropriate research methodology and suitable data collection instruments. Vigilant interactive theory, which is essentially an open attentive group discussion enabled by the use of EI trait behaviours, is shown to support the propensity of the group to produce higher quality decision-making processes. Petrides and Furnham (2000) identified two distinct forms, EI ability and EI trait. Martins et al.'s (2010) meta-analysis shows that EI trait has been reported to have a stronger relation to adaptive outcomes than EI ability. These findings suggest the proposition that EI trait has a greater likelihood of facilitating better quality decisions. Thus, the second part of the survey used in the case study operationalised vigilant interactive theory by developing questions which were based on the ASX Corporate Governance Council's Principles. These consist of 8 principles which recommend corporate governance practices for ASX listed entities.

This research is primarily focussed on the second of these principles which refers to the "Structure the board to be effective and add value" and relates to the structure of the board being such as to enable the board "to have the skills, commitment and knowledge…to enable it to discharge its duties effectively and to add value" (ASX 2019). This research also incorporated aspects of the AICD's "Good Governance Principles for Non-for-Profit Organisations: Principle 6 Board Effectiveness" (Australian Institute of Company Directors 2018). Vigilant interactions enable effective governance practices to operate so that the themes may be identified (Hirokawa and Rost 1992). For example, purposeful and vigilant discussion, and examination of the options available relating to a decision to be made by the board, often leads to a more optimal decision. This was considered to be the case by the directors as vigilant discussion and constructive debate enabled differing views to be shared by the directors and enabled directors to be persuaded otherwise by the supporting views and alternative options presented.

A qualitative coding process within NVivo to identify and synthesise themes and common observations was employed. NVivo was also used to help provide further explanations of the correlative relationships observed from the detailed narratives provided by the directors (Kemp 2011). The directors discussed their perception of the effectiveness of the board and its culture, as well as how these aspects were impacted, and changed, during a six-month study period (Yin 2012). Both the surveys conducted before and after the interviews were designed based on the specific principles of various authoritative sources.[1] Likert-type scaled ratings were adopted for close-ended responses (Institute of Community Directors Australia 2018).

## 4. Data Analysis

A quantitative analysis was conducted first. The survey focussed on the directors' self-assessment of how the effectiveness of their boards' decision-making processes was modified over the course of the case study. The purpose of conducting this analysis was to further explore and understand whether the directors perceived that the effectiveness of the chair had a positive impact on their views of the effectiveness of their boards in decision-making. A single factor ANOVA analysis of all the 18 participating directors in

the case study was conducted for both 'pre' and 'post' surveys (i.e., completed before and after interviews).

Two skills of the behavioural competencies identified by Kiel et al. (2012) are collaboration and understanding of effective decision-making processes. Firstly, an ANOVA analysis was conducted for the Chair and non-Chair groups, as content and reliability validity checks, and produced results showing no statistically significant differences. This grouping was able to provide evidence that there were no significant differences between these two skills for both the sender's (the chair) perspective as well as the receiver's (the non-chair) perspective. Secondly, a 'pre' and 'post' validity test was performed using an ANOVA analysis. The results in Table 1 support that there were no statistically significant results in the group of directors between the 'pre' and 'post' results ($F = 0.028$; $p = 0.870$ for chairs and $F = 0.076$; $p = 0.786$ for non-chairs) for the standardised board decision-making effectiveness results over the six-month case study period.

**Table 1.** Comparison of population averages for board decision-making effectiveness.

|  | Role | Board Decision-Making Effectiveness (Standardised) |
|---|---|---|
| | All | 5.67 |
| Pre interview survey (averages) | Non-chair | 5.72 |
| | Chair | 5.61 |
| | All | 5.63 |
| Post interview survey (averages) | Non-chair | 5.69 |
| | Chair | 5.57 |

The descriptive statistics showed a decreased effectiveness of the board's decision-making processes trend, and this may be due to a moderating effect of the COVID-19 pandemic that the Australian financial services entities are currently navigating.

## 5. Findings from Two Surveys and Interviews

The following discussions are based on the observations of two survey responses and a qualitative analysis of the responses from interviewees.

### 5.1. Observations on Key Themes from the Surveys

The discussion in this section focusses on the observations made from the highly rated questions in the surveys that related to the effectiveness of the board decision-making processes. A number of interesting observations were recorded which highlighted that the directors considered their boards to be very good and effective in the following:

- Facilitating relevant, robust discussion to enable all views to be shared;
- Considering a variety of potential solutions, negative and positive aspects before making a decision;
- Cultivating diversity of opinion in board discussions;
- Taking the time to consider the real issue at hand;
- Expressing confidence in sharing points of view where it is open to the possibility of two different ideas;
- The effectiveness of the chair in facilitating and engaging differing views in board decisions;
- The chair makes a significant contribution to the effective running of board decisions and the way forward.

The directors' confidence factors were high, feeling that they could share their points of view even where it was open to the possibility of different competing ideas in a board decision-making process, increased on overall average during the six-month period from a 6.0556 rating to a 6.1111 rating. This positive direction of their confidence serves to

demonstrate they have the view that they were able to facilitate rigorous discussion which may have resulted in more optimal decision-making and outcomes, consistent with vigilant interactive theory (Hirokawa and Rost 1992).

*5.2. Analysis of Interview Data*

Unanimously, all directors interviewed agreed and acknowledged the value in rigorous discussion amongst the fellow directors at the board table as part of good governance practices and the decision-making processes of their boards:

> . . .*if you have a diverse range of people and views which discuss matters in a robust way, you'll get a better discussion rather than one opinion which is always the decision being made. It is always something that enables a better decision.* (Director 8)

A theme that emanated consistently amongst the directors was the pivotal role that the board chair or committee chair played in facilitating the diversity of views amongst the collective group of directors to enable a richness in discussion and better quality outcomes in the decision-making process:

> *A lot of it depends on the chair and how the chair can draw out others' perceptions, how to manage the strong minded and dominating directors on boards.* (Director 12)

How the chair and directors achieved this in terms of behavioural traits was described in varied ways by the participants. According to one director, often the language used had common themes of "mutual respect", "listening", "self-awareness", and "*open to being persuaded otherwise*".

Synonyms of vigilance, particularly "rigorous" and "robust" discussion, were two common expressions used to confirm the value in the board governance practices of achieving better decision-making as a board:

> *It works differently with different boards. The board papers you get provide 90% of the assessment of the risks and strategy for most matters. Robust discussion is very useful and the chair facilitates this and suggests the way forward for a decision to be approved.* (Director 11)

Some of the directors shared further observations that they hoped broader governance practices of other boards and emotional intelligence traits of the broader director community would continue to be brought even further to the forefront of how boards operate:

> *I hope that COVID-19 changes a lot of directors and their appetite for risk, tolerance and hopefully even their trust as a board, given we're having to work together over extensive amounts of time through this period.* (Director 13)

This observation by director 13 highlights one moderating influence of COVID-19 on the board decision process has been a greater appreciation of risk tolerance and collaboration amongst board members. A participant director of a large Australian Financial Services Institution stated.

Also, COVID-19 has led to the necessity for directors to distinguish between critical and non-critical projects in decision-making processes to focus the board on critical matters, in the short term during the pandemic. One participant made the following comment.

> *Enablement of great autonomy of management within the framework of board governance and oversight. This included distilling decisions to focus on key critical issues at the board level and organisational culture through COVID-19, bring forward decisions that could have been 10 years in the making which were recognised as critical to pivot and delay non-necessary projects during the pandemic.* (Director 11)

A heightened awareness of the board members of their company's stakeholders, and not just its shareholders, has been the moderating effect of the pandemic on the board's decision-making process. The directors considered that emotional intelligence traits and consciousness as a board, including greater awareness and foresight of the impacts of COVID-19 on their market, stakeholders, community, and branding was a key undervalued

trait in the traditional sense of the key board skills of directors. COVID-19 had heightened the awareness of this and its value in decision making.

In addition to the market impact of COVID-19, regulatory changes at State, Territory, and Federal levels are other impositions on the board's governance and its decision-making processes. Directors noted that well-being was very important as part of their emotional intelligence in the effectiveness of board decision-making processes, to be able to engage and make clear, sound, and prudent decisions during economic and health crisis that brought constant changes during 'black swan' events, such as the Coronavirus pandemic. COVID-19 brought on significant market movements, domestic economic changes, and a vast tranche of Federal and state legislative and regulatory change to grapple with.

While there was an awareness of the behavioural dynamics within the board's culture and operations as a result of the Coronavirus pandemic, the impact of this external imposition was in parallel with the previously imposed need to change behavioural culture report by the Royal Commission. These sentiments were reflected in the following two comments by a director of a large Australian Financial Services Institution:

> *The Coronavirus pandemic in Australia (and globally) recalled a similar heightened awareness as did the Royal Commission in the Australian financial services sector for directors in relation to the importance behavioural dynamics, empathy and emotional intelligence played on the culture and of operation of boards.*

> *The Royal Commission [and the COVID-19 pandemic] has given a lot to pause and think about. Is the customer really being treated fairly? Boards are now being challenged with—well how do you know? I think there's been a shift, there's growing awareness in how you treat your stakeholders. You've got to go deeper and understand your business not relying on making assumptions.* (Director 7)

Another board director of a large Australian financial institution summarised the COVID-19 impacts of the board decision-making process in the following manner:

> *The Australian financial services sector had gone through so much over the last 25 years (including through the GFC) without too much going wrong. Therefore, overtime, naturally with human nature, board and management become less risk adverse in decision-making as a result. Risk and non-risks were assessed annually in decision-making which included factoring in pandemics.*

> *Now, in light of the COVID-19 pandemic—the unthinkable is thinkable in decision-making of businesses. It is anticipated in light of COVID-19, in the context of boards and management: risk aversion will decrease in decision-making.*

> *It's not easy to have the same kinds of decisions and discussions when you're not in the room, you can't read the body language, the emotion of other directors and gage whether now is a good time to bring up an issue or not. The most important parts of board meetings and decisions happen in the tea rooms and hallways.*

> *New directors have great challenges with getting integrated into the board or business without having built up the relationships of a business over years face to face. This has a big impact on decision-making in the COVID-19 environment.* (Director 7)

It was noted during the interviews that from this global pandemic, the directors hoped the boards of Australia's large financial services institutions would be more resilient and trusting as a result of this experience.

## 6. Conclusions, Limitations, and Implications

The possibility of this research being moderated by uncontrollable events and data not necessarily gathered in a static and stable context, but rather under differing circumstances, has created a challenging backdrop to this research. Therefore, directors could be considered to have been constrained in their views and to have maintained a level of conservativism, or healthy scepticism. They considered their feelings to be positive about the future success

and growth of the organisations they governed. They also had a positive attitude to their ability to make successful or effective decisions that would lead to good quality outcomes.

Personal attributes such as self-control reflect an individual's ability to rationality and healthily manage, as well as balance their impulses, external pressures or stresses. This factor had a low positive correlative relationship with the directors' self-assessments of the effectiveness of their board's decision making and perhaps was not as impactful on the effectiveness of the board, demonstrating that the group in the decision-making process was important. These results support the existence of an effectiveness of the board decision-making, which suggests that the effectiveness depends on a well-functioning group dynamic of the directors. This suggests that the directors are likely to have a positive mindset for growth, and an instilled level of confidence in the board's decision-making processes. The boards of Australian financial services institutions are well developed, structured, and mature; therefore, this does lay the foundation for a methodical way of making decisions and facilitating the open vigilant discussion, with confidence, that the directors' views and opposing ideas will be heard and respected.

The Coronavirus global pandemic has significantly impacted Australia's financial services industry and the economy. The results from the surveys could have perhaps been distorted by moderating factors of the Coronavirus pandemic that occurred during the same time as the research study between January and June 2020. For example, perhaps the Australian financial services institutions had many competing factors to grapple with at the board room, and the board was not as effective with decision-making under pressure, during testing times, or was placed into differing circumstances. For example, board meetings were conducted outside the normal hours of the day via Zoom or Skype remotely. Also, the technological environment had distorted the collective boards' experience in the board meetings and their ability to engage emotionally with each other as effectively in the decision-making process as they did in person.

The directors were curious, had an open mindset, respected the responsibilities of their roles as directors, and considered that the culture of their boards was collegiate with a strong currency of trust amongst their fellow board directors. They considered that emotional intelligence did have a positive impact on the quality of decision-making and had referred to their own personal experiences of learning to use emotional intelligence traits over their careers which enabled them to navigate through better experiences in board deliberations to achieve more preferred and optimal outcomes as a result.

This research has found that there was a currency of trust which enabled directors to develop a deeper awareness of their emotional engagement with other directors, and this facilitated greater effectiveness in board decision-making processes. It has also highlighted that the boards of Australian financial services organisations were going on a journey. This was a journey of maturity in developing a conscious awareness of how the skills and diversity of personalities of directors in the composition of the board had a positive impact and influence on the effectiveness and success of strategic decisions. The directors often reflected that a lot of the achievements of their boards as the collective group were able to be acknowledged by looking back with the benefit of hindsight.

The key themes emanating from the research are the following:

(1) boards were going on a journey of developing skills and diversifying personalities of directors, and the composition of the board had a positive impact and influence on the effectiveness and success of strategic decisions.

(2) The directors acknowledged during their interviews that a lot of this was achieved by their boards when looking back in hindsight.

(3) The main theme for this study that arose from the interviews in relation to the effectiveness of the board decision-making process was that the effectiveness in the chair was pivotal in facilitating robust discussion and enabling a structure approach to the decision-making process of the board.

The normal limitation of limited generalisations of findings for research using a qualitative design methodology for 18 participants applies (Vasileiou et al. 2018). Also,

the Coronavirus global pandemic highlighted some of the limitations to cross-sectional research, but the results do identify opportunities to investigate the effect of EI traits in future research in both developing and developed countries during recurring pandemic and post-pandemic conditions. Two other themes emanating from this study can be identified by future research. Firstly, the influence of the emotional intelligence of the chair and their ability to facilitate open and structured discussion in board decision-making processes was critical in the effectiveness of the board. Secondly, that the business currency of trust which was facilitated through the use of the emotional intelligence skills of directors enabled board decision-making processes to take place with robust discussion and a respected environment where directors could express their views and persuade each other otherwise.

These two potential themes may provide a level of consistency assumed in the board decision-making practices experienced by the participating directors in the research case study from the different boards of large financial services institutions in Australia or other countries. From a quantitative research design perspective, there are likely to have been mediating factors involved, which may be of use for further research to explore their effects. For example, whether certain aspects of emotional intelligence personality traits produced better results than others, which were not explored in this research.

The opportunity for further exploring EI traits used by directors, or extended to include other senior executives or owners, on the journey as EI traits mature for the collective and individuals. This future research should provide a better understanding and explore how the boards or management achieve these strategies, compared to the directors' experiences, observations, and views on this process in the current study.

**Author Contributions:** Conceptualization, J.H. and J.S.; methodology, J.H.; software, J.H.; validation, C.B. and J.H.; formal analysis, J.H.; investigation, J.H.; data curation, J.H., J.S., G.J., and C.B.; writing—original draft preparation, J.H.; writing—review and editing, J.S., G.J. and C.B.; visualization, J.S. and G.J.; supervision, J.S., G.J. and C.B.; project administration, J.H. All authors have read and agreed to the published version of the manuscript.

**Funding:** This research received no external funding.

**Conflicts of Interest:** The authors declare no conflict of interest.

## Note

1    The ASX Corporate Governance Council's '*Corporate Governance Principles and Recommendations: Principle 2 Structure the Board to Add Value*" (ASX Corporate Governance Council 2014) '*Corporate Governance Principles and Recommendations*', and the AICD's '*Good Governance Principles for Non-for-Profit Organisations: Principle 6 Board Effectiveness*' (Australian Institute of Company Directors 2018).

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
