# Peer review of "Did Emotional Intelligence Traits Mitigate COVID-19 Uncertainty Effects on Financial Institutions’ Board Decision-Making Process?"

_jrfm, doi:10.3390/jrfm17030106_

Round 1

Reviewer 1 Report

Comments and Suggestions for Authors

Authors examine the "COVID-19 effect on financial institutions’ board decision-making process" using the mixed methods, interviews, and Nvivo for textual analysis. The work seems interesting, and some of the observations are about the sense of self and EI, the board's decision-making ability in the financial industry. But, the authors fail to investigate the issue scientifically.

1. There has been very limited literature and theoretical aspects covered in the current study. Section Introductions lack the basic motivation of the study and rationale.

2. Authors fail to identify research questions and hypotheses. 

3. There is no discussion about the theory or prominent literature that talks about the board's decision-making process during uncertainty.

4. Authors need to borrow some essential studies and frameworks to deal with the board characteristics and decision making process

5. The author fails to justify their sampling process/interview and the number of participants from the board.

6. Author presents some statistics using ANOVA, e.g., There was no statistically significant results in the group of directors between the ‘pre’ and  ‘post’ result (F = 0.028; p = 0.870 for chairs and F = 0.076; p = 0.786 for non-chairs)..... it seems that F-stats appears insignificant. what are those hypotheses, and could not be proved? How authors can generalize their findings based on weak results.

Comments on the Quality of English Language

--

Author Response

Thank you for your feedback, our responses to each of your comments is attached.

Reviewer 2 Report

Comments and Suggestions for Authors This is an interesting piece of work. However, the following corrections should be made to the paper for enhancing the quality of the paper.

1. Research motivation describes what motivates researchers and what they would like
to achieve as a result of their studies. Therefore, the author can add a paragraph on research motivation in the introduction.
2. The author has written a research gap in the introduction section. But the same can be explained in detail with supportive relevant literature published in the last 10 years.

3. Authors could explain the implications of study for society, regulators and policymakers. 4. Author/s should explain in the Introduction how this study would be useful for the global audience with supporting literature.

5. Literature Review should be extensive and include more supporting literature published in the last 10 years.

6. Research methods could be motivated further showing the use of a particular methods. How literature review was done? Inclusion - exclusion criteria.

Comments on the Quality of English Language

Its fine. Proofreading required.

Author Response

(The authors gave the same response as above.)

Reviewer 3 Report

Comments and Suggestions for Authors

The authors are recommended to revise the Introduction by including all the necessary elements: Research relevance, scientific gap, research goal, methods applied, a brief description of the main findings.

The research goal should be formulated.

The structure of the paper should be improved. Now, it is too fragmented. It is not recommended to develop a separate section that includes just one paragraph (for instance, 2.2, 2.3). 

The literature analysis is vague. The theoretical background has been supported by only 18 information sources.

The authors mentioned NVivo as one of the data processing tools; hovewer, there are no outputs generated by NVivo included.

Findings should be described in details.

Author Response

(The authors gave the same response as above.)

Round 2

Reviewer 1 Report

Comments and Suggestions for Authors

The authors fail to address the research question scientifically, and there are still many flaws, a lack of theoretical underpinning, and a review of prominent studies.

Author Response

Reviewer 1 has provided feedback of a very general nature “The authors fail to address the research question scientifically, and there are still many flaws, a lack of theoretical underpinning, and a review of prominent studies.”  We cannot be guided by the limited information provided by reviewer

We wish to thank you 1 for the time taken to read our paper and ask for specific description of the work needed; especially when the other two reviewers are satisfy with the revised paper and they do not require any further revisions before the paper is published.

As we cannot provide specific revisions to your general feedback, our decision is to response to each of your comments with our defense of the extensive revision to the revised paper that 2 of the 3 reviewers have accepted as satisfactory and requesting more specific details from you so we can attend to ryour specific requests.

The author team

Reviewer 2 Report

Comments and Suggestions for Authors

Authors has presented a nice work including all my suggestions.

Author Response

Many thanks for your review and comments.

The author team.

Reviewer 3 Report

Comments and Suggestions for Authors

No additional comments

Author Response

(The authors gave the same response as above.)
